# Deep Reinforcement Learning Microgrid Optimization Strategy Considering Priority Flexible Demand Side

**DOI:** 10.3390/s22062256

**Published:** 2022-03-14

**Authors:** Jinsong Sang, Hongbin Sun, Lei Kou

**Affiliations:** 1Changchun Institute of Technology, School of Electrical Engineering, Changchun 130012, China; sjs15855726241@163.com; 2National and Local Joint Engineering Research Center for Smart Distribution Network Measurement, Control and Safe Operation Technology, Changchun 130012, China; 3Institute of Oceanographic Instrumentation, Qilu University of Technology (Shandong Academy of Sciences), Qingdao 266075, China; koulei1991@qlu.edu.cn

**Keywords:** microgrid, energy storage, flexible load, reinforcement learning, deep learning, energy optimization

## Abstract

As an efficient way to integrate multiple distributed energy resources (DERs) and the user side, a microgrid is mainly faced with the problems of small-scale volatility, uncertainty, intermittency and demand-side uncertainty of DERs. The traditional microgrid has a single form and cannot meet the flexible energy dispatch between the complex demand side and the microgrid. In response to this problem, the overall environment of wind power, thermostatically controlled loads (TCLs), energy storage systems (ESSs), price-responsive loads and the main grid is proposed. Secondly, the centralized control of the microgrid operation is convenient for the control of the reactive power and voltage of the distributed power supply and the adjustment of the grid frequency. However, there is a problem in that the flexible loads aggregate and generate peaks during the electricity price valley. The existing research takes into account the power constraints of the microgrid and fails to ensure a sufficient supply of electric energy for a single flexible load. This paper considers the response priority of each unit component of TCLs and ESSs on the basis of the overall environment operation of the microgrid so as to ensure the power supply of the flexible load of the microgrid and save the power input cost to the greatest extent. Finally, the simulation optimization of the environment can be expressed as a Markov decision process (MDP) process. It combines two stages of offline and online operations in the training process. The addition of multiple threads with the lack of historical data learning leads to low learning efficiency. The asynchronous advantage actor–critic (Memory A3C, M-A3C) with the experience replay pool memory library is added to solve the data correlation and nonstatic distribution problems during training. The multithreaded working feature of M-A3C can efficiently learn the resource priority allocation on the demand side of the microgrid and improve the flexible scheduling of the demand side of the microgrid, which greatly reduces the input cost. Comparison of the researched cost optimization results with the results obtained with the proximal policy optimization (PPO) algorithm reveals that the proposed algorithm has better performance in terms of convergence and optimization economics.

## 1. Introduction

With the development of power systems for a variety of distributed energy sources, the traditional energy management system (EMS) is also expected to develop into a new form called an integrated energy management system (IEMS). The construction of IEMSs can optimize energy supply for microgrids and provide new decisions for optimal demand-side scheduling [1,2]. The reasonable optimal scheduling of demand-side energy is the most direct way for energy supply to play a significant role. The energy management of microgrids mainly faces the problem of optimal scheduling. In the case of large-scale distributed energy resource (DER) intervention, the large-scale volatility of DERs and the randomness of demand-side response have brought many problems of optimal allocation and scheduling. With the development of artificial intelligence machine learning, deep reinforcement learning can effectively solve the above problems.

From the perspective of the microgrid model, reference [3] can effectively manage energy distribution for distributed power generation, energy storage systems and price response users without day-ahead forecasts; reference [4] constructs wind power and a distributed energy storage system in which users achieve a certain degree of Nash equilibrium without pre-experiment. Reference [5] considers demand response and energy storage systems. In the day-ahead scheduling stage, according to the short-term forecast information of wind power, a day-ahead scheduling scheme is formulated with the minimum system power generation cost as the objective function, and the day-ahead scheduling plan of the system is revised. Reference [6] establishes a model for the thermostatically controlled loads (TCLs) on the user’s demand side and uses reinforcement learning for optimal control of energy-saving scheduling. Reference [7] simulates the microgrid environment of battery energy storage combined with hydrogen storage devices and uses the deep Q-network (DQN) reinforcement learning method to complete energy scheduling optimization. Components such as DERs and ESSs are considered in [8] to form a small campus microgrid combining power conversion and users, and a hierarchical reinforcement learning network is used for optimal scheduling of microgrid energy. In [9], the microgrid model takes into account distributed power sources, users and electric vehicles to reasonably coordinate the load state to optimize the economical operation of the power grid. Reference [10] considers price response and user participation in a microgrid model with wind and solar output and uses intraday consumption for scheduling optimization. Reference [11] pointed out that the real-time price signal formulated by the reinforcement learning algorithm (DRL) is used as the dominant signal to manage the operation of the microgrid, and at the same time, the deep neural network is used to learn the behavior of the user side of the microgrid.

From the aspect of microgrid optimization algorithm, references [12,13] used mixed-integer linear programming, cooperative game and alternating direction multiplier methods and at the same time cooperated with different time scales such as intraday rolling optimization and real-time adjustment. The performance of the algorithm for the microgrid energy optimization strategy was further improved. Reference [14] considers the DQN algorithm to learn the real-time scheduling strategy of the microgrid, discretizes the battery energy storage action, greatly reduces the range of optional actions and improves the efficiency of the learning strategy. Reference [6] uses an improved DQN algorithm to carry out the scheduling optimization of the microgrid composite model of energy storage and battery. This algorithm uses the double-layer learning network of DQN to reduce the correlation of network update parameters and improve the learning efficiency. Reference [15] shows that the cost of the double DQN strategy is higher than that of the continuous action reinforcement learning algorithm through the simulation case of real-time control of a residential multienergy system. Therefore, from the perspective of the economy of management strategy, it is necessary to introduce continuous action-type reinforcement learning algorithms. Reference [16] proposed a method of optimizing comprehensive energy economic dispatch by using the deep deterministic policy gradient (DDPG) algorithm for renewable energy considering the time-varying characteristics of the load. This method addresses environmental policy learning on continuous state actions. Aiming at the uncertainty of renewable energy and load, reference [15] achieves robust optimization based on mathematical programming (MP). Reference [17] integrates stochastic optimization for day-ahead optimized energy management; however, although these factors are considered comprehensively, there is still a deviation from the actual operation. Reference [18] constructed a multiagent and multiobjective DRL architecture but did not consider the limitation of training ability. Reference [19] adopted the asynchronous advantage actor–critic (A3C) algorithm to effectively manage microgrid units such as energy storage and power generation. This algorithm greatly improves the speed of training and convergence.

The above existing research on the optimal dispatch of microgrid energy, on the one hand, includes ESSs, distributed energy resources, controllable loads, TCLs, power conversion, price response loads and the main grid from the perspective of the microgrid model. However, there are few studies on dispatch optimization of these combined microgrids in current research. On the other hand, from the perspective of microgrid optimization algorithms, the existing research [13,14,15] optimization algorithms include MP, Q-learning, DQN and DDPG. Although they can solve the problem of high-dimensional decision-making of various loads under nonconvex problems, there are still many drawbacks. For example, deep Q-learning cannot select reasonable actions on consecutive actions. The DQN algorithm uses the gradient algorithm to effectively solve such problems, but the update mode of the algorithm adopts the mode of round update, which greatly reduces the learning efficiency. Later, scholars proposed the DDPG algorithm, which based on the actor–critic algorithm can not only efficiently solve the energy management problem of continuous action space, but also improve the learning efficiency. However, the deterministic strategy is not conducive to the exploration of actions, and easily falls into a local optimum.

Inspired by the previous studies, this paper not only combines distributed energy resources, controllable loads, TCLs, power conversion, ESSs, price response loads and the main grid from the perspective of the microgrid model, but also adopts priority to TCLs, ESSs and price response loads to achieve control, increased flexibility in demand response and improved energy utilization. From the perspective of microgrid optimization algorithm, combined with the existing research, the experience playback pool M-A3C is introduced on the basis of the A3C algorithm. This algorithm solves the effective management and training of high dimensions and the speed of convergence and achieves higher model performance and convergence of optimal policies.

The major contributions of this study are as follows:This paper establishes a microgrid model, including battery packs, incentive response user groups and related typical components that can participate in main grid power dispatch.In the flexible scheduling of battery packs and the participation of the main network, considering the optimal scheduling strategy of daytime scheduling and real-time scheduling, a method is proposed that considers the priority of the energy storage system participating in the energy interaction between the main network and the microgrid. When mapping the above uncertainty into the Markov decision process (MDP) in the setting of the reward function, the minimization of the operating cost of the microgrid system is regarded as the reward value maximization form of reinforcement learning.A memory library A3C algorithm is proposed to efficiently utilize data and reduce the interaction time between agent and environment during training. The M-A3C algorithm is compared with other algorithms (DQN, PPO, double DQN), and it is verified that the strategy optimization ability of the algorithm is better than that of the other algorithms in the same environment.

## 2. Microgrid Structure and Equipment Model

As shown in Figure 1, the microgrid model structure given in this paper mainly considers the infrastructure requirements with an independent supply and demand environment. The power of the microgrid is generated by its own wind turbines, while at the same time interacting with the main grid in both directions. The power of the main grid and the power of the microgrid are purchased and sold through the implementation and fluctuation of market electricity prices. Figure 1 mainly shows the physical layer, information layer and control layer. For example, the state of charge of the battery, the purchase and sale of electricity in the microgrid and the price of electricity can all be transmitted through the information layer. The control layer mainly includes the charging and discharging of the battery controlled by the microgrid, the purchase and sale of the residential price response load power and the main grid power. It is generally divided into three types of control: thermostatically controlled load (TCL) switching control, energy storage system (ESS) discharge and main network electricity trading.

### 2.1. Objective Function

The goal of microgrid demand-side energy optimization is to achieve the lowest-cost-oriented microgrid energy management decision under the time-scale scheduling of rolling operation within the day. Considering the cost of DER power generation, the depreciation of energy-storage-optimized batteries and the energy dispatch cost of the main grid participation, the mathematical expression of the operating cost of the system is as follows:(1)$$Y=\min(C_{E}+C_{ESS}+C_{G})$$
(2)$$C_{G}=\sum_{t=1}^{T}c_{b,t}|\delta_{t}{|}^{+}-c_{s,t}|-\delta_{t}{|}^{+}$$
where $C_{E}$ is the power generation cost of DER; $C_{ESS}$ is the depreciation cost of charging and discharging of the battery in the energy storage system; *T* is the operating time period within a day; $\delta_{t}$ is the amount of electricity exchanged between the microgrid and the main grid at time *t*; when $\delta_{t}$$\delta_{t}$ > 0, it means buying electricity from the main grid, otherwise it means selling electricity to the main grid; $c_{b,t}$ is the price of purchasing electricity from the main grid at time *t*; and $c_{s,t}$ means buying electricity at the main grid price of electricity.

The microgrid energy optimization operation constraints include power balance constraints, main grid energy supply constraints and related equipment operation constraints.

#### 2.1.1. Power Balance Constraints

The electric power balance constraints of the microgrid and the thermal power balance constraints of TCLs at time t in the day are as follows:(3)$$\delta_{t}+P_{wind}(t)-P_{ESS}(t)+P_{load}(t)=0$$
(4)$$\sum_{i=1}^{N}P_{TCL,i}(t)=H_{load}(t)$$
where $P_{wind}(t)$, $P_{ESS}(t)$ and $P_{load}(t)$ are the wind power, energy storage battery power and residential load power at time *t*, respectively; N represents the number of TCLs, $P_{TCL,i}(t)$ represents the power of the *i*-th TCL at time *t*, and $H_{load}(t)$ is the thermal power of the microgrid at time *t*.

#### 2.1.2. Interaction Power Constraints between Microgrid and Main Grid

Based on the stable operation of the demand side of the microgrid, there are upper and lower constraints on the power interaction between the main grid and the microgrid:(5)$$P_{grid}^{\min}\leq P_{grid}(t)\leq P_{grid}^{\max}$$
where $P_{grid}^{\min}$ and $P_{grid}^{\max}$ are the upper and lower limits of the interaction power between the microgrid and the main grid, respectively.

### 2.2. Model of Each Component of the Microgrid

This paper uses the real wind energy production data of a wind farm, and the data volume is real-time data of 240 days and 5760 h. These data are shared with the microgrid in real time, as the power of the wind farm is sent to the local microgrid. $P_{t}^{g_{t}}$ is the output power of wind power at time *t*, $\beta^{wind}$ is the conversion efficiency of wind power and $P_{t}^{wind}$ is the maximum power of wind power; then,
(6)$$P_{t}^{wind}=\beta^{wind}*P_{t}^{g_{t}}$$

In the ESSs model, ESSs serve as short-term reserves, and their charge–discharge response signals are sent by the battery. ESSs in microgrids can maintain the balance between supply and demand of microgrid energy. $S_{bat}(t)$ represents the real-time energy of the battery; $S_{bat}(t-1)$ is the energy before charging and discharging; $P_{bat}^{cha}$ and $P_{bat}^{dis}$ are the charging and discharging powers, respectively; and $\eta_{b}$ is the charging and discharging efficiency. Then,
(7)$$S_{bat}(t)=\left\{ \begin{array}{l} S_{bat}(t-1)+\eta_{b}\int P_{bat}^{cha}dt\\ S_{bat}(t-1)-\int \frac{P_{bat}^{dis}}{\eta_{b}}dt \end{array}\right.$$

The charge–discharge response behavior for a single ESS is provided and emitted by the battery. For example, when the battery sends a discharge signal, the EMS accepts its stored power and distributes it to the microgrid load, and when there is energy surplus, the excess energy will be sold to the main grid. When the charging signal is released, if the microgrid energy allocated by EMS cannot be satisfied, the rest will be automatically supplied by the main grid. In addition, for the battery, the damage of charging and discharging should also be considered. We use $B_{soc}(t)$ to represent the real-time state of charge, so the battery has a certain range constraint:(8)$$B_{soc}^{\min}\leq B_{soc}(t)\leq B_{soc}^{\max}$$
(9)$$S_{bat,\min}\leq S_{bat}(t)\leq S_{bat.\max}$$
(10)$$0\leq P_{dis,t}\leq P_{dis,\max}$$
(11)$$0\leq P_{cha,t}\leq P_{cha,\max}$$
(12)$$B_{soc}(t)=\frac{S_{bat}(t)}{S_{bat,\max}}$$
where $B_{soc}^{\min}$ and $B_{soc}^{\max}$ are the upper and lower limits of the battery energy storage state; $S_{bat,\min}$ and $S_{bat.\max}$ are the minimum and maximum battery capacities, respectively; $P_{dis,\max}$ and $P_{cha,\max}$ are the maximum powers of battery discharge and charge, respectively.

The TCL model has a thermostatic control cluster and includes various temperature control loads, and energy conservation provides a great possibility of flexible scheduling. It is assumed that each residential household is equipped with refrigerators, air conditioners, heat pumps, water heaters, etc. These components are uniformly controlled by the TCL manager, and the real-time temperature may be expressed as follows:(13)$$T_{in}^{t}=H_{air}*(T_{out}^{t-1}-T_{in}^{t-1})+T_{b}^{\Delta t}+P_{tcl}*S_{b,t}*H_{in}^{t-1}$$
where $T_{in}^{t}$ and $T_{out}^{t-1}$ are the real-time indoor temperature and the temperature at the previous moment, respectively; $H_{air}$ and $H_{in}^{t-1}$ represent the air heat in the building and the heating inside the building at the previous moment; $T_{b}^{\Delta t}$ is the temperature change in the building; and $P_{tcl}$ is the average power of TCL. For switches $S_{b,t}$, the following action relationships exist:(14)$$S_{b,i}^{t}=\left\{ \begin{array}{l} T_{in,t}^{t}>T_{\max,i}\to Cooling\\ T_{in,t}^{t}<T_{\min,i}\to Heating\\ T_{\min,i}<T_{in,t}^{t}<T_{\max,i}\to Static \end{array}\right.$$
where $T_{\max,i}$ and $T_{\min,i}$ represent the upper and lower limits of the *i*-th TC temperature and $T_{in,t}^{t}$ represents the real-time control temperature of *i* at time *t*.

For many TCLs, there must always be a certain priority level, which we use $SOC_{i}^{t}$ to denote:(15)$$SOC_{i}^{t}=\frac{T_{in,t}^{t}-T_{\min,i}}{T_{\max,i}-T_{\min,i}}$$

So for each TCL, we will only charge the cost of generating electricity for the energy consumed. In order to ensure user comfort, temperature verification and switch operation are carried out according to the priority EMS.

In the residential price response model, the microgrid cannot directly participate in the control of residential load energy consumption. Assuming that the user’s electricity consumption behavior follows the price rolling law within the day, we can change the real-time response of residential load through the intervention of price. Therefore, each house is given two constraints that affect power, load flow and response compensation, which can be represented by ρ and τ. The load flow ρ represents the proportion of load loss and increase under the condition of intraday price rolling, and ρ∈[0,1]; the response compensation τ represents the cost of the microgrid for users to reduce electricity consumption due to price fluctuations. Then, the residential i power load $L_{i}^{t}$ at time t can be modeled as follows:(16)$$L_{i}^{t}=L_{sum,b}-\rho *L_{sum,b}*\mu_{t}+\tau$$
(17)$$\mu_{t}\left\{ \begin{array}{l} >0\to High-price\\ <0\to Low-price \end{array}\right.$$
where $L_{sum,b}$ represents the basic load of the house at time *t*. $\mu_{t}$ represents the price level at this time; when it is positive, it represents a high price level, and when it is negative, it represents a low price level, for the load caused by intraday price fluctuations. The churn compensation cost τ can be expressed as follows:(18)$$\tau =\sum_{t=0}^{T-1}\alpha *\rho *L_{sum,b}*\mu_{t}$$
where α∈[0,1] is a constant, which is the execution probability of the compensation load under a certain time step; in order to balance the energy supply and demand balance between the main network and users, a pricing mechanism should be designed with reference to market prices. The intraday average price $M_{avg}$ of the microgrid should be below 10% of the market price on that day, which can be expressed as $\frac{M_{avg}-M_{market}}{M_{market}}<10\%$. $M_{market}$ represents the market price; the price can be described as follows:(19)$$P_{t}=M_{market}+\mu_{t}*w$$
where *w* represents the price increase or decrease constant.

### 2.3. Interaction Mechanism between Main Network and Microgrid

The main grid can supply power to the microgrid immediately when its energy is insufficient, or the main grid can receive excess power when the microgrid has excess energy. Transactions between the main grid and the microgrid take place in real-time using price up $P_{t}^{U}$ and price down $P_{t}^{D}$. $\delta_{t}$ represents buying or selling to the main grid; positive values represent energy purchased, and negative values represent energy sold.

## 3. Microgrid Management Reinforcement Learning Scheme

The microgrid energy management in this paper is actually a sequential decision-making problem in an uncertain environment. Adopting a model-free or data-driven approach can better adapt to environmental uncertainties.

The reinforcement learning algorithm M-A3C achieves parallel operation in multiple environments by using threads, allowing multiple agents with substructures to update the parameters in the main structure in these parallel environments at the same time. In parallel, the agents do not interfere with each other. After the environment interacts with a certain amount of data, the gradient is updated in its own thread. However, these gradients do not update the neural network in its own thread but update the neural network of the main structure. Therefore, the correlation of the algorithm update is reduced, and the convergence is greatly improved.

### 3.1. Markov Decision Problem

The deep reinforcement learning microgrid energy scheduling problem is modeled as an MDP.

#### 3.1.1. State Space

In this microgrid model, the state information in the environment includes the state of charge of TCLs $SOC_{t}$, residential load $P_{t}$, temperature $T_{t}$, real-time power generation $G_{t}$, state of charge of the energy storage system $BSC_{t}$, market price $C_{t}$ and daily consumption load value $L_{t}$; therefore, the state space at time t is expressed as follows:(20)$$S=\left\{L_{t},C_{t},BSC_{t},SOC_{t},P_{t},T_{t},G_{t},L_{t}\right\}$$

#### 3.1.2. Action Space

Each time the agent obtains the state information, it makes a selection in the action space according to the strategy. For the actions of many devices in the microgrid, please refer to the components in the first section. There are mainly five possible actions for the residential load based on the price, so as to formulate the price level $A_{load}$; the constant temperature control load has four actions $A_{tcl}$ to ensure that each TCL action is carried out through their respective priorities for the energy storage equipment at t. The action at a time is the input $A_{D}$ and output $A_{E}$ of energy. Therefore, the discretized action has 80 possibilities, which can be expressed as follows:(21)$$A=\left\{A_{load},A_{tcl},A_{D},A_{E}\right\}$$

#### 3.1.3. Reward Function

In reinforcement learning, the reward value is used to map the quality of the strategy, and the management goal of the microgrid is to minimize the cost. On the one hand, this paper regards the minimization of the system operating cost of the microgrid as a form of reward value maximization in reinforcement learning. On the other hand, in order to ensure that the algorithm framework has good convergence, a profit function is also added as another part of the reward. It can be expressed as follows:(22)$$R=C_{profile}-C_{\cos t}$$
where $C_{profile}$ and $C_{\cos t}$ represent the revenue of the microgrid and the cost of the microgrid, respectively;
(23)$$C_{profile}=P_{t}*(\sum_{n=1}^{N}L_{i}^{t}+\sum_{i=1}^{I}P_{tcl}*S_{b,t}^{t})+P_{t}^{D}*|\delta_{t}{|}^{+}$$
(24)$$C_{\cos t}=C_{ESS}+P_{t}^{U}*|-\delta_{t}{|}^{+}$$
where $C_{ESS}$ represents the depreciated cost of the battery.

### 3.2. M-A3C Network Structure

The A3C algorithm is a multithreaded online learning algorithm based on the actor–critic (AC) framework, which combines the evaluation method of policy and value. It is also an improvement and upgrade of the AC algorithm. The A3C algorithm model combines two parts, actor and critic. Actors are responsible for generating actions and participating in real-time interactions with the environment, while critics evaluate and direct the actions of actors. The input state of the actor environment is *S*, and the action policy in the current environment is output π(S). The input state of the critic environment is S, and the output is the evaluation value v(S) of the current state S, and v(S) represents the value of the expected mean state S. The A3C algorithm uses the difference between the action value function and the state value function (optimization function) as the criterion for evaluating the critic, combined with the value accumulation of the N part, as shown in Equation (25), where A(S,t) is the optimization function, representing the value of the current state *S*, where γ is the attenuation factor.
(25)$$A(S,t)=R_{t}+\gamma R_{t+1}+\cdots \gamma^{n-1}R_{t+n-1}+\gamma^{n}v(S^{\prime })-v(S)$$

The critic network in the A3C algorithm uses the TD-error value of δ and uses the mean square error as the loss function to update the parameters of its own network parameters of ω. The calculation formula is shown in Equation (26).
(26)$$\begin{array}{l} \delta =R+\gamma v(S^{\prime })-v(S)\\ loss={\sum \left(R+\gamma v(S^{\prime })-v(S,\omega )\right)}^{2} \end{array}$$

The evaluation of the critic network directly affects the update of the actor network parameters, and the policy entropy term is added to the policy loss function. Then the parameter θ of the actor network is updated as shown in Equation:(27)$$\theta =\theta +\alpha \nabla_{\theta }\log\pi (s_{t},a_{t})A(S,t)+c\nabla H(\pi (S_{t},\theta ))$$

The objective function of the model under maximum entropy can be expressed as follows [20]:(28)$$\pi^{*}=\arg\underset{\theta }{\max}E_{(s_{t},a_{t})\to \rho \theta }\left\{\sum r(s_{t},a_{t})+\tau H[\theta (•|s_{t})]\right\}$$
where *H* is the entropy term taken in state $s_{t}$ and τ represents the weight of exploration ability. The goal is to ensure that the entropy of the resulting action is as large as possible in a state $s_{t}$. While maximizing the reward of the agent, the model should also make the agent have more space for exploration. The optimal target of the agent is represented by $Q_{*}(s_{t},a_{t})$; that is, the optimal value represents the maximum cumulative reward that can be obtained by $s_{t}$, and $Q_{*}(s_{t},a_{t})$ can be obtained through the Bellman maximum. The optimal equation calculation is as follows:(29)$$Q_{*}(s_{t},a_{t})=E[r_{t+1}+\lambda \max a_{t+1}Q(s_{t+1},a_{t+1})|s_{t},a_{t}]$$
where λ is the reward discount factor.

The actor makes an action based on the current state of the simulated environment, and the critic gives a reward for the actor’s performance based on the state and action [21]. Actors adjust their strategies based on critics’ rewards. Finally, the actor learns the optimal policy from the environment, and the critic also gives the optimal reward value.

In the A3C algorithm, the actor network and the critic network can generally take the form of a fusion network and a separation network for practical tasks. Based on the reinforcement learning A3C algorithm, this paper introduces an experience playback pool in the reference deep Q-learning algorithm. The experience playback mechanism obtains historical data. Relevant information breaks the correlation between data, and the reuse of experience also increases the efficiency of data usage. The M-A3C interactive environment and training process are shown in Figure 2.

Each substructure network accepts the state of charge from TCL, residential load, temperature, real-time power generation, state of charge of the energy storage system, market price and load value of daily consumption from the environment. In each time period *t*, each substructure evaluates and updates the state, and at the same time, the learned strategies are unified by the main structure to gather experience and finally update the strategy. The experience of each episode in the training process will be stored in the experience pool to improve the speed of update and the efficiency of learning.

## 4. Results

### 4.1. Basic Data

In order to verify the effectiveness of the improved M-A3C algorithm based on Algorithm 1 proposed in this paper and the energy scheduling of the proposed model, Appendix A lists the parameters of each component of the microgrid. Twenty-four hours of the day will be used as the time period for microgrid energy scheduling optimization. The main power grid, residential houses, energy storage systems and thermostatic control systems are set up in the environment, and the residential houses and thermostatic control systems are presented in a cluster.
**Algorithm 1.** Pseudocode for each thread learning in A3CAssume globally shared parameter vectors θv and θ global shared count *T* = 0Assuming thread-specific parameter vectors θv′ and θ′Initialize thread steps *t*←1Repeat   Reset gradient: dθv←0 and dθ←0;    Synchronized thread parameters: θv′=θv and θ′=θ;    tstart=t;   Get the state at this time *t*
st;   Repeat    According to strategy π(at|st;θ′), get the state at at this time *t*    At time *t* reward value rt and new state st+1    t←t+1    T←T+1    if st or t−tstart=tmax terminate   R={0,  st→terminateV(st,θv′)st, st→non−terminating and starting from the last state    For i∈{t−1,⋯⋯tstart}    R←ri+γR;    Calculate about the gradient θ′: dθ←dθ+∇θ′logπ(ai|si;θ′)(R−V(si;θ′))   Calculate about the gradient θv′: dθv+∂(R−V(si;θ′))2/∂θv′End.Use dθ to perform asynchronous updates of θ and θv and dθv;only T>Tmax


The simulation environment in this paper considers the addition of wind turbines and takes into account the uncertainty of wind power fluctuations. However, in the end, the generator model was not used, but the power generation data of the Finnish wind power plant in [22] were used, and the data resolution was 1 h. Figure 3 shows the sample wind power data, and Figure 4 shows the basic data of components in the microgrid for one day. In the selection of the hyperparameters of the algorithm M-A3C, after many running comparisons, this paper sets the experience sample playback capacity in the M-A3C algorithm to 500; the smallest batch is 200, the update batch is 100 and the learning rate is 0.001. The training iteration is 1, and ε−decay is set to 0.00005.

On the basis of basic data, the microgrid performs energy dispatching optimization among components. The agent directly controls the TCLs and residential loads. In the microgrid environment, the number of TCLs is set to 150. The agent allocates a certain amount of energy to the TCLs and determines the priority according to each SOC. Reasonable switching action is performed to stabilize the temperature in the range of 18 to 28 °C. The residential load is set to 150, and each load has two parameters. Due to the fluctuation of the electricity market price, the increase or decrease in the load will be compensated accordingly in the later stage. For the energy surplus and energy shortage of the energy storage system, the main network participates in the interactive response for a reasonable distribution of energy.

### 4.2. Analysis of Results

The microgrid environment is trained by M-A3C and PPO, and the convergence results are shown in Figure 5. Each epoch corresponds to each day of training in the environment. At the beginning of training, the agent does not obtain the corresponding environmental information, so it obtains a relatively small reward value after executing the scheduling decision. However, with the accumulation of the training process, the reward value is also continuously accumulated, and the agent continues to learn the scheduling strategy, so the overall trend of the reward value increases and finally tends to converge. From Figure 5, the M-A3C and PPO are compared and analyzed, and the two converge in about 30 episodes; that is, after training for 30 days, both can explore the optimal scheduling strategy, but the final reward value for convergence of M-A3C is obviously higher than that of PPO. M-A3C has a certain fluctuation in the final convergence, but it has a high reward value, which means that the profit is relatively large; meanwhile, the performance of PPO is limited.

In order to illustrate the effectiveness of the M-A3C algorithm in decision-making, a certain day from the 40th to the 120th in Figure 5 is selected for analysis at an hourly optimization interval. As can be seen in Figure 6, at 11:00–12:00, the market electricity selling price is the highest and the microgrid sells a lot of energy. While the market electricity selling price is relatively low at 22:00–24:00, the microgrid buys a large amount of energy. At the same time, as shown in Figure 7, the ESS will buy the storage at 5:00–12:00, and the amount of charge gradually increases and tends to be saturated, which can effectively relieve the power supply pressure of peak electricity, but after 15:00, the microgrid system is in rapid energy consumption. In this stage, the charge of the ESS decreases significantly, which indicates that the ESS continues to supply energy for the microgrid and ensures the stability of the load power consumption. In summary, under the influence of real-time electricity price, M-A3C realizes the optimal dispatching strategy of microgrid energy, and the microgrid obtains better economy.

Finally, comparing the daily electricity production charts of the microgrid in Figure 6 and Figure 8, it can be seen that the large-scale distribution and consumption of energy in a day basically correspond to the low market price of electricity in a day, which can greatly reduce the cost of investment. As shown in Figure 9, the distribution of TCL energy in a day is around 00:00–8:00 to ensure the supply of subsequent energy in the day; a large amount of energy is purchased in the main network to ensure sufficient allocation of early loads, which can be very stable during peak electricity prices.

### 4.3. Comparison of Algorithms

In order to further verify the superiority of the M-A3C algorithm in microgrid energy scheduling, the input dimension of the neural network is set to a 7-dimensional state vector, and the output is the state-relative action value $s_{t}\to a_{t}$; there is a hidden full connection in the neural network layer, the number of neurons is 100 and the activation function is ReLU. We compared the PPO and M-A3C algorithms in the same microgrid environment and selected the cost of each day for 10 days from the 40th to 120th day of training and the retailer’s cost for these 10 days for comparative analysis.

As shown in Figure 10, the cost of M-A3C for 7 of the 10 days showed a good low cost due to PPO. The data show that the standard deviation of the 10-day cost for M-A3C is 0.526, the standard deviation for PPO is 0.681 and the standard deviation for the retailer’s 10-day cost is 0.73. The cost output under the reinforcement learning algorithm achieves a cost reduction of 37.8% compared to the retailer’s planned output. Compared with the PPO algorithm, the cost input is reduced by 34%. It can be seen that M-A3C can efficiently handle the energy dispatching optimization problem of the microgrid. Figure 11 shows the optimization results of the main microgrid energy dispatching under the PPO algorithm. The data in the figure are selected for 24 h on the same day as in Figure 6. It can be seen from the comparison chart that under the same intraday electricity market buying and selling price, the microgrid sells more electricity on that day. Purchase prices in the electricity market peaked at 11:00–12:00. During the 22:00–24:00 and 00:00–6:00 periods of the electricity price valley, the microgrid also purchased a larger amount of electricity. Therefore, compared with the M-A3C optimization under the same conditions within 24 h, the PPO-optimized microgrid energy interaction output is relatively small, and the corresponding purchase volume is larger. Although they are all at the bottom of the electricity price in the electricity market, the cost is far greater than the optimization result of M-A3C, which is also reflected in Figure 10.

Table 1 further compares the reward value of DQN and double DQN under different experience pools and batch sizes of training batches. In the same environment, the test uses different algorithms to optimize the policy for the environment objective function. The results show that the final scores of PPO and M-A3C are greater than those of DQN and double DQN in training batches of 200 and 500 with the same size of experience replay pool. The final score of M-A3C is due to the PPO algorithm, and it can be seen that adding M-A3C is the best choice. Table 2 compares the score levels of the offline–online-based strategy PPO algorithm and the improved M-A3C when parameter 5 is changed. From Table 2, on the one hand, it can be clearly seen that the PPO score based on offline and online strategies is significantly weaker than the score of M-A3C at the same ε−decay during the learning process. On the other hand, the PPO algorithm’s scoring ability fluctuates greatly under different ε−decay, while M-A3C can show stable scoring ability under different changes in ε−decay and obtain a more stable solution. Therefore, it can be seen from the synthesis that the improved M-A3C is better than DQN, double DQN, etc. in the algorithm’s scoring ability. At the same time, it has a good generalization ability for parameter changes and shows better performance.

## 5. Discussion

In this paper, a model-free reinforcement learning method is proposed for the uncertain real-time scheduling problem of multiple loads in microgrids. Reinforcement learning establishes the MDP decision-making component. The proposed improved M-A3C algorithm based on actor–critic is a relatively complete method for solving the microgrid energy optimization problem proposed in the article. The following conclusions can be drawn regarding the use of this method in microgrid optimization:

(1) The application of reinforcement learning M-A3C to the proposed system with the integration of TCL cluster, residential load, energy storage system and external power grid results in good adaptability.

(2) The algorithm training results show that the used M-A3C algorithm shows better convergence than the PPO algorithm and obtains higher profit rewards. Through multithreaded synchronous training, M-A3C can not only improve the training speed, but also update certain parameters from each thread at the same time and return the parameters to the experience pool and the main network at the same time for collection and optimization, and the experience pool is fully updated. The new and old strategies of parameters are used, which reduces the correlation between strategies and improves the convergence speed. Although the PPO algorithm combines offline–online sampling, the sampling of the PPO algorithm has a certain correlation and cannot fully learn random strategies to improve optimal scheduling.

(3) In the analysis and comparison of the results of the M-A3C algorithm, it is concluded that the intraday optimization cost is better than that of the retailer, and the cost reduction ratio in 10 days reaches 0.36. In the future, further discussion and research are needed on the generalization ability of the model and how to use more advanced algorithms and more flexible loads to participate in collaborative optimization.

## Figures and Tables

**Figure 1 sensors-22-02256-f001:**
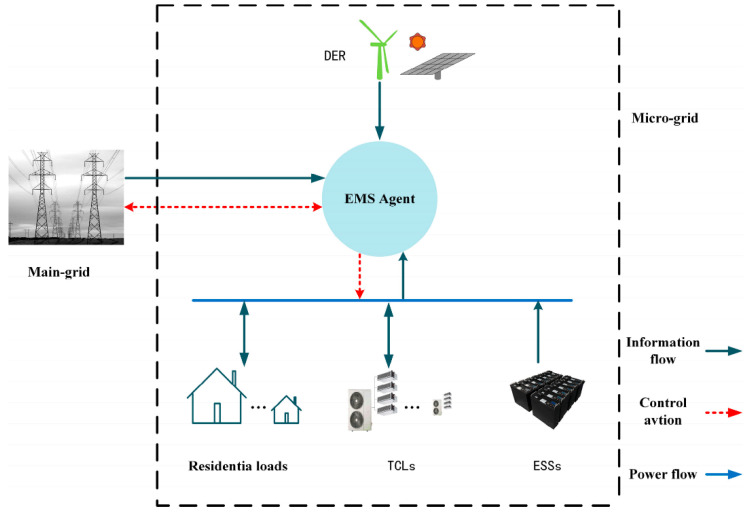
Microgrid architecture.

**Figure 2 sensors-22-02256-f002:**
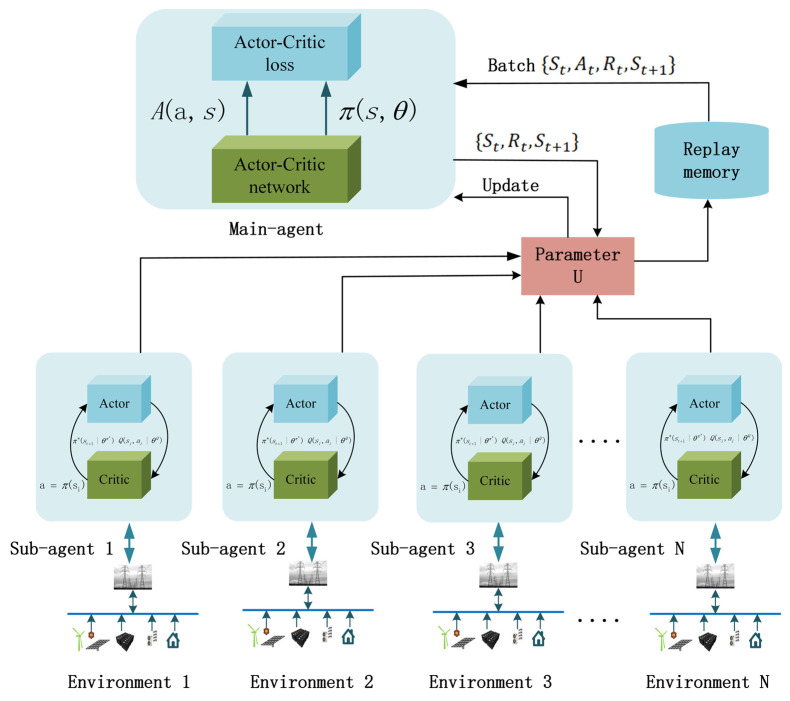
M-A3C microgrid management structure.

**Figure 3 sensors-22-02256-f003:**
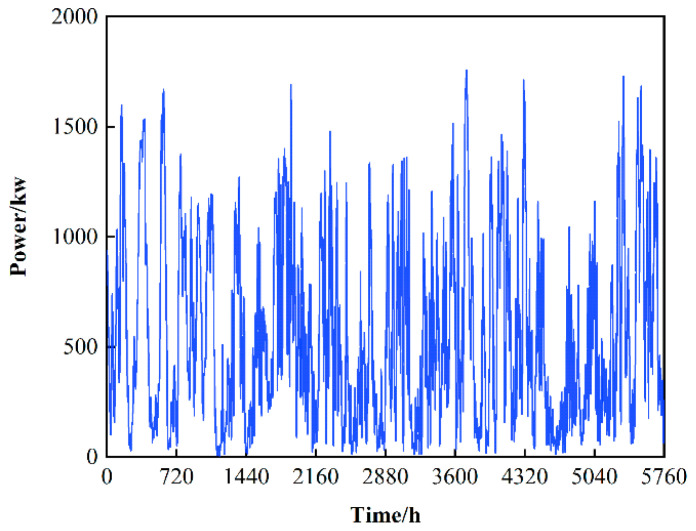
Microgrid wind power generation curve.

**Figure 4 sensors-22-02256-f004:**
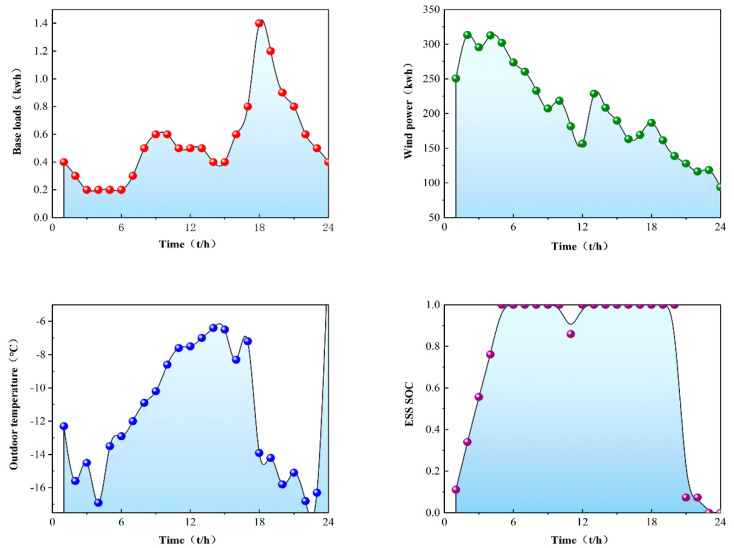
Initial state of microgrid environment components.

**Figure 5 sensors-22-02256-f005:**
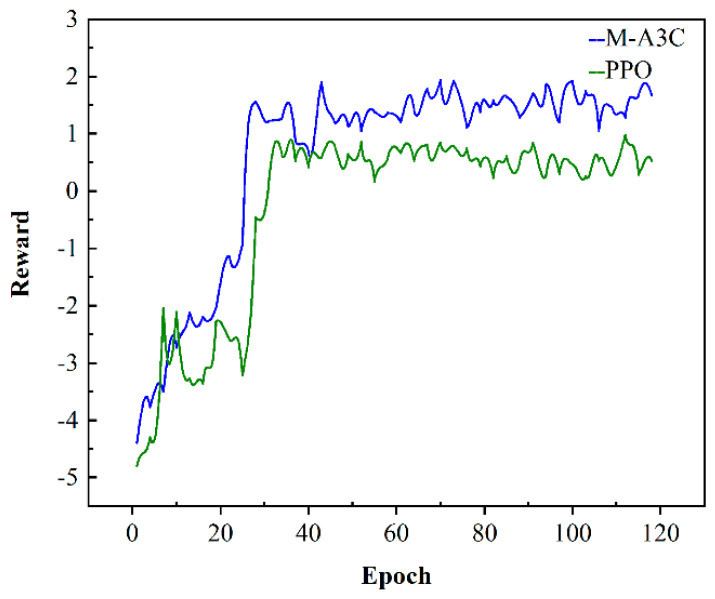
Cumulative reward curves for M-A3C and PPO training.

**Figure 6 sensors-22-02256-f006:**
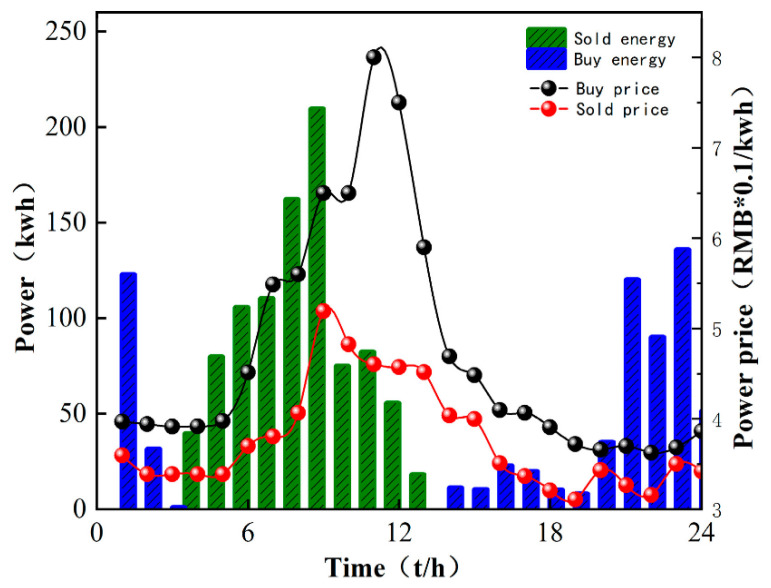
Microgrid and main grid energy interaction.

**Figure 7 sensors-22-02256-f007:**
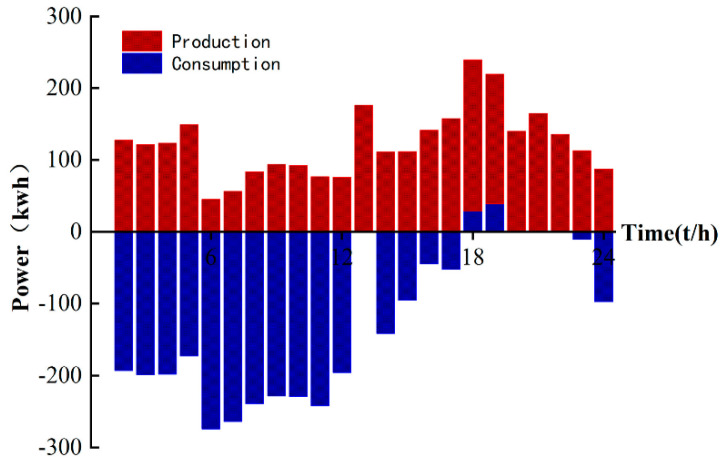
Microgrid and TCL energy dispatch.

**Figure 8 sensors-22-02256-f008:**
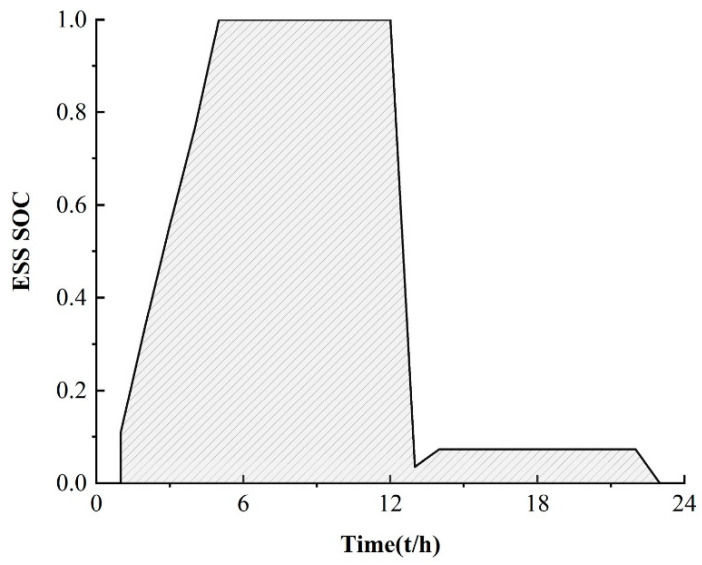
ESS daily charge.

**Figure 9 sensors-22-02256-f009:**
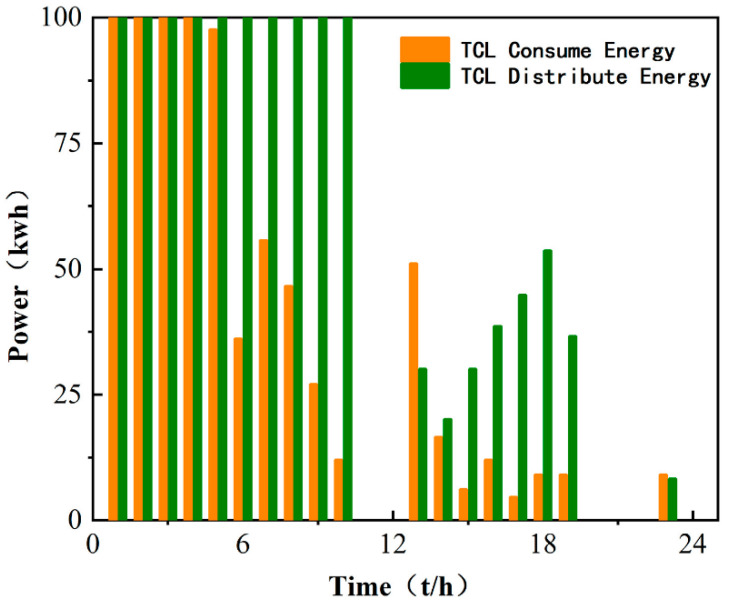
Daily distribution and consumption of TCL.

**Figure 10 sensors-22-02256-f010:**
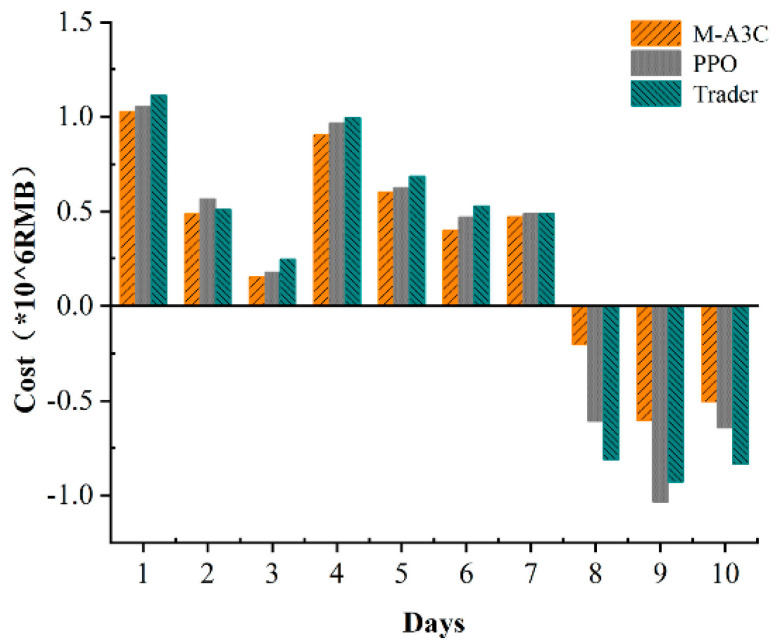
A3C and PPO optimization performance comparison.

**Figure 11 sensors-22-02256-f011:**
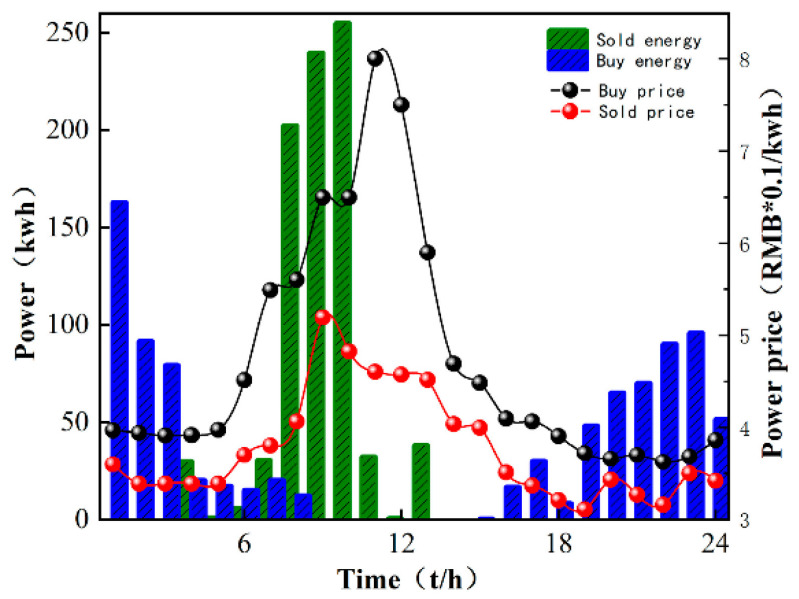
Energy interaction between PPO microgrid and main grid.

**Table 1 sensors-22-02256-t001:** Comparison of reinforcement learning algorithms based on different experience pools.

$M_{\max}$	Batch Size	DQN	Double DQN	PPO	M-A3C
300	200	0.080	0.166	0.213	0.246
	500	0.140	0.154	0.310	0.243
700	200	−0.270	0.162	0.11	0.245
	500	0.123	0.151	0.09	0.250

**Table 2 sensors-22-02256-t002:** Comparison of reinforcement learning algorithms based on different ε−decay.

ε−decay	PPO	M-A3C
$$5\times 10^{-5}$$	−0.921	0.106
$$1\times 10^{-5}$$	−0.396	−0.082
$$1\times 10^{-4}$$	−0.152	0.286

## Data Availability

Not applicable.

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
