# Peer review of "Deep Reinforcement Learning Microgrid Optimization Strategy Considering Priority Flexible Demand Side"

_sensors, 2022, doi:10.3390/s22062256_

Round 1

Reviewer 1 Report

This paper introduces a reinforcement learning based algorithm for micro-grid optimization, and compares it against some of the existing algorithms in literature. Below are my comments:

  • The paper requires extensive editing and English corrections. Below are some of the mistakes I caught:
    • Line 351, “…1 is set to 0.00005”. Was there supposed to be a parameter in place of “1”?
    • Line 374, “Scope;” Was there supposed to be a sentence involving the word “Scope”?
    • Line 375-376: “According to the fluctuation of the market price, compensation according to the increase or decrease of the load;” The sentence has no verbs.
    • Line 152-154: “The price and the price at which electricity is sold to the main grid, which is subject to market price fluctuations” The sentence seems to have been misplaced from the previous sentence?
    • Line 188-189: “And distributed to the micro-grid load, the difference there is provided by the main grid, and vice versa” The sentence seems to be incomplete.
    • Line 313: “The probability that the actor modifies the selected behavior based on the critic's score.” The sentence is incomplete. Was it supposed to be “is based” instead of “based”?
    • Line 43: IMES should be IEMS?
  • Line 150: It is not clear how delta_t represents the electricity exchanged at time t. Is it power, energy or price?
  • Please spell out A3C the first time it is used in the paper. Is it “Asynchronous Advantage Actor Critic”?
  • Line 381, the paper mentions “Episode” and in Figure 5, the label of the x-axis is “Epoch”. Are they the same? If so, for clarity it is suggested to use the same word.
  • Line 152: what is the difference between c_b_t and c_s_t? The paper says they are the electricity purchased from the main grid at time t but the sentence seems to be incomplete. Does c_s_t mean the electricity sold to the grid and c_b_t the electricity bought from the grid? If so, please clarify in the sentence that introduces them. Also do they represent, power, energy or price?
  • The paper evaluates the performance of the proposed algorithm on a specific micro-grid configuration with a specific set of data for wind power generation curve. It is suggested that the performance of the proposed algorithm is evaluated on multiple different configurations or at least different wind-power curves so that the results on the superiority of the proposed algorithm against other algorithms are more conclusive and generalizable.
  • Figure 5 shows the training of M-A3C algorithm and PPO over 120 consecutive days (around four months). Then, Figures 6-9 show the behavior of M-A3C algorithm on a single day. Is this a representative day after the 120 days of training or it is among the days inside the 120 days? Similarly figure 10 compares the performance of A3C and PPO algorithm over 10 days. Are these 10 days after the 120 training days shown in figure 5 or part of it? Also are these 10 consecutive days or some randomly selected days? Please clarify these points in the paper.

Author Response

Thank you for taking the time and effort to review our work. Your valuable comments have greatly helped us improve the paper significantly. Based on the comments, we revised our paper. Modifications are highlighted in the revised manuscript, and our responses are summarized in the annexes below.

Reviewer 2 Report

The article “Deep reinforcement learning micro-grid optimization strategy 2 considering priority flexible demand side” is interesting and contemporary. I have some minor comments before final publication.

  1. Line 48: DER appeared for the first time. Authors should use the whole word for the terms which appear for the first time and then use the abbreviations. An abstract is different from the main text. The authors should define the terms again in the introduction section.
  2. Line 65, 71, 77, 74, 61, etc.: It is better to write that “The authors consider -------- (9). The reference style is vague. Make changes throughout the introduction section.
  3. Line 332: Results instead of the result
  4. Section 4.3. This section is quite weak. More explanation should be added.
  5. Section 5. The discussion should be separate from conclusions.
    The reference list is inconsistent. The authors should follow the MDPI style.

Author Response

(The authors gave the same response as above.)

Reviewer 3 Report

The paper addresses a topic of interest.

1. Most of the presented relations are common for the thematic. 

2. The authors should better highlight the improving they proposed to the algorithm. 

3. Having a look on the Reference list, I did not identify any paper of the authors. Is this their first paper in this field?

Author Response

(The authors gave the same response as above.)

Reviewer 4 Report

The reviewed manuscript concerns deep reinforcement learning micro-grid optimization strategy. I have the following critical remarks concerning this work:

  1. In the Abstract is “Markov Decision Process (MDP) process”.
  2. The introduction should be improved. In the Introduction is written, e.g. “literature [3] can”, “Reference [5] establishes”, “in Reference [7] integrates”. The way of description of references should be changed. In the Introduction (lines 41-43), is “energy management system (Energy Management System, EMS) is also expected to develop into a new form called integrated energy management system (Integration Energy Management System, IEMS)”. On p. 2 (line 88) is “are available in the literature.”, but the number of the reference isn’t given.
  3. On p. 1 (line 43) is “IMES”, but should be “IEMS”.
  4. For the first time, an abbreviation with the full name should be given in the text (in each of three sections: the abstract; the main text; the first figure or table), and next, only an abbreviation could be used, e.g., in the Abstract is “distributed energy resources (DER)”, but should be “distributed energy resources (DERs)”, on p. 2 (line 47) is “DER”, but should be given with the full name, on p. 3 (line 117), is “the Markov decision process, but should be “the Markov Decision Process (MDP)” and on p. 7 (line 267) is “Markov decision process”, but should be “MDP”. All abbreviations should be explained.
  5. On p. 4 (line 143), is “DES”. I suppose that should be “DER”.
  6. In scientific papers, it doesn’t use “we” (e.g. p. 5, 6, 10) but sentences in passive voice.
  7. On p. 9 is “the figure below”, but should be “the Figure 2”.
  8. On p. 9 (line 331), is “The M-A3C algorithm accepted by each environment can be described as follows:”, below is “Pseudo-code for each thread learning in A3C:” and below and on the next page are the calculations given - the strange form – it is the table? The fonts and the font size are different.
  9. On p. 3 (line 131), is “The figure”, on p. 10 (line 336), is “in the following figure” – which numbers of figures?
  10. Table 1 isn’t cited in the main text, but in the manuscript, there is Table 1. In the manuscript, there are two Figures 8.
  11. Tables 2 and 3 should be described in the manuscript.
  12. On p. 11 (line 346), is “Figure 3 below” but should be “Figure 3” – Fig. 3 isn’t below, but on the next page. On p. 13 (line 381) is, “Figure 5 below, but should be “Figure 5”.
  13. References should be unified according to the Guide for Authors. The name of the title of the journal should be given in abbreviation. The last names of all authors of a publication should be given. Only the first letter of the author's first name should be given.
  14. Editorial mistakes should be corrected e.g.

- on. p. 1 (line 14), is “DER. problem.”, but should be “DER problem”,

- on p. 2 (line 61) is “ESSs ,” but should be “ESSs,”,

- on p. 2 (line 74) is “algorithm The scheduling” but should be “algorithm. The scheduling”,

- on p. 6 (line 229), is “Modeled” but should be “modelled”,

- on p. 8 (line 281), is “at t The action”, but should be “at t. The action”,

- on p. 11 (line 351), is “0.00005;”, but should be “0.00005.”,

- the numbers from lines 342, 485 and 487 should be deleted.

  1. English should be carefully checked, e.g.

- in many places in the manuscript, there are sentences with repeated the same words, e.g. on p. 1 (lines 12-14) – “DER”, (lines 14-15 and 32, p. 2, lines 61-62, p. 5, lines 181-182, p. 7, 248-249) – “micro-grid”, p. 3 (lines 114-115) – “scheduling”, (lines 123-124), p. 14 (lines 451-452), p. 16 (line 496) – “algorithm”, p. 6 (line 201) – “TCLs”,

- on p. 3 (line 100), “The content studied in this paper” sound not good,

- on p. 14 (lines 442-443) is “Figure 7 with Figure 6”, but should be “Figures 6 and 7”.

According to mentioned above remarks, I suggest that in this paper, the major revision is needed before publication in Sensors.

Further comments:

1. Chapter 2.2 is not clear. Some of symbols should be defined. In the fist part of this sentence "is the output power of wind power at time t, is the conversion efficiency of wind power, and is the maximum power of wind power, then:" is something missing.

2. Data given in Table 1 should be presened in other form.

Author Response

(The authors gave the same response as above.)

Round 2

Reviewer 1 Report

The authors have responded to my comments. I have no more comments.

Author Response

We appreciate your time and efforts in reviewing our work. Your valuable comments significantly help us improve the paper significantly.

Reviewer 4 Report

The some of the reviewer's comments were taken into account. I have the following critical remarks concerning this work:

  1. In the Abstract is “Markov Decision Process (MDP) process”.
  2. The introduction should be improved. In the Introduction is written, e.g. “literature [3] can”, “Reference [5] establishes”, “in Reference [7]”. The way of description of references should be changed. In the Introduction (lines 41-43), is “energy management system (Energy Management System, EMS) is also expected to develop into a new form called integrated energy management system (Integration Energy Management System, IEMS)”. On p. 3 (line 100) is “are available in the literature.”, but the number of the reference isn’t given.
  3. For the first time, an abbreviation with the full name should be given in the text (in each of three sections: the abstract; the main text; the first figure or table), and next, only an abbreviation could be used, e.g., on p. 2 (lines 47-48) is “distributed energy resources (DER)”, but should be “distributed energy resources (DERs)”, on p. 3 (line 129), is “a markov decision process, but should be “the Markov Decision Process (MDP)” and on p. 7 (line 281), is “Markov decision process”, but should be “MDP”. All abbreviations should be explained.
  4. In scientific papers, it doesn’t use “we” (e.g. p. 11, 15, 16) but sentences in passive voice.
  5. On p. 11 (line 372), is “in the following table 1”, but should be “in Table 1”.
  6. On p. 9 (lines 346-349), is “Actors choose their actions based on probability, and Critic judges their scores based on their actions. The probability that the actor modifies the selected behavior based on the critic's score. Further, Actor learns the optimal value function from the environment, while Critic learns the state-action function.” – it isn’t clear.
  7. The order of the figures should be changed.
  8. Data given in Table 1 should be presented in other form.
  9. References should be unified according to the Guide for Authors, e.g. the name of the title of the journal should be given in abbreviation, the last names of all authors of a publication should be given.
  10. Editorial mistakes should be corrected e.g.

- on p. 2 (line 57), is “loads(TCLs)” but should be “loads (TCLs)”,

- on p. 2 (line 75), is “adjustment, The” but should be “adjustment. The”,

- on p. 2 (lines 79-80), is “algorithm The scheduling” but should be “algorithm. The scheduling”,

- on p. 3 (line 57), is “research[13-15]” but should be “research [13-15]”,

- on p. 5 (line 193), is “micr-ogrids” but should be “micro-grids”,

- on p. 5 (line 202), is “grid . When” but should be “grid. When”,

- on p. 5 (line 204), is “grid In” but should be “grid. In”,

- on p. 6 (line 101), is “Modelled” but should be “modelled”,

- on p. 12 (line 386), is “0.001, The” but should be “0.001. The”,

- on p. 13 (line 412), is “stage; For” but should be “stage. For”,

- on. p. 13 (lines 425-426), is “is Ob-viously” but should be “is ob-viously”,

- on p. 15 (line 492), is “analysis” but should be “analysis.”,

- the text on p. 16 should be justified.

  1. English should be carefully checked, e.g.

- “micro-grid” or “microgrid”?

- in many places in the manuscript, there are sentences with repeated the same words, e.g. “DER”, “micro-grid”, “scheduling”, “algorithm”,

- on p. 3 (line 100), “The content studied in this paper” sound not good,

- on p. 15 (lines 477-478), is “Figure 6 and Figure 7”, but should be “Figures 6 and 7”,

- on. p. 16 (line 537), is “when the parameter 5 is changed”.

According to mentioned above remarks, I suggest that in this paper, the minor revision is needed before publication in Sensors.

Author Response

Thank you for taking the time and effort to review our work. Your valuable comments have greatly helped us improve the paper significantly. The changes to the article have been replied in the attachment。
